# Knowledge and attitudes of deaf persons towards safe abortion services in Ghana

**Wisdom Kwadwo Mprah[1], Juventus Duorinaah[2], Maxwell Peprah Opoku[3]\*, William Nketsia[4], Michael Amponteng[4]**

1 Centre for Disability and Rehabilitation Studies, Department of Health Promotion and Disability Studies, Kwame Nkrumah University of Science and Technology, Kumasi, Ghana, 2 Juventus Duorinaah, Ghana National Association of the Deaf, Accra-North, Ghana, 3 Special Education Department, United Arab Emirates University, Al-Ain, United Arab Emirates, 4 School of Education, Western Sydney University, Sydney, Australia

\* Maxwell.p@uaeu.ac.ae

**Data Availability Statement:** Data cannot be shared publicly because of ethics restrictions. Data are available from the Kwame Nkrumah University of Science and Technology/Ghana Health Service (contact via chrpe.knust.kath@gmail.comHannah.

## Abstract

### Background

Deafness refers to partial or total loss of hearing, which, if not appropriately accommodated, may interfere with day-to-day living experiences. Deaf people encountered challenges in their efforts to access essential services, such as health care. While some attention has been given to general access to reproductive health services, less research has focused on the experiences of deaf women and girls when accessing safe abortion services. With unsafe abortion being a major cause of maternal deaths among women in developing countries, this study attempted to explore the perception of deaf women and girls in Ghana towards safe abortion services.

### Objective

The main aim of this study was to understand the perception and awareness safe abortion services among deaf women and girls in Ghana. In doing this, the contributors towards unsafe abortion practices among deaf women and girls were gathered.

### Method

Penchansky and Thomas' accessibility to health care theory availability, accessibility, accommodation/adequacy, affordability, and acceptability guides this study. A semi-structured interview guide based on components of the theory was used for data collection from 60 deaf persons.

### Results

The components of the theory were used as a priori themes that guided the data analysis. The results showed challenges associated with the indicators of health access. For instance, in terms of availability, it was revealed that deaf women had little knowledge about existing laws on safe abortion in Ghana. In relation to acceptability, deaf women were highly

Frimpong@hru-ghs.org) for researchers who meet the criteria for access to confidential data.

**Funding:** This work was funded by a grant from the Safe Abortion Action Fund (Empowered Deaf Women for Healthy Life Grant, EMDEWOHL) awarded to all authors.fv

opposed to abortion for cultural and religious reasons. However, there was consensus that safe abortion could be conducted under certain conditions.

## Conclusion

The results of the study have implications for policymaking aimed at attaining equitable access to reproductive health care for deaf women. The need for policymakers to expedite public education and incorporate the needs of deaf women in reproductive health policies, as well as other study implications, are discussed.

## Introduction

Globally, induced abortion is one of the most sensitive and debated topics that pervades the cultural, social, moral, religious, and legal facets of life [1, 2]. An induced abortion is a procedure undertaken intentionally to terminate a pregnancy before the fetus is capable of extra-uterine life [1]. Recently, Centers for Disease Control and Prevention [3] defined induced abortion as an "an intervention performed by a licensed clinician e.g., a physician, nurse-midwife, nurse practitioner, physician assistant within the limits of state regulations, that is intended to terminate a suspected or known ongoing intrauterine pregnancy and that does not result in a live birth." [https://www.cdc.gov/reproductivehealth/data_stats/abortion.htm]. Induced abortion is categorized into three kinds: safe, less safe, and least safe. Safe induced abortions are undertaken by medical practitioners with requisite skills in an environment that conforms to the highest medical standards, and they carry minimal risks [1]. While less safe abortion refers to termination of pregnancy by trained professionals who do not follow the right procedure or right method, least safe abortion is the termination of pregnancy by an individual without specialized training or the ability to follow standard medical procedures [1]. In many advanced countries, safe abortion is legalized and made accessible to women. However, abortion has been criminalized in many sub-Saharan African countries, thus restricting access to safe abortion-related services [2, 4–10]. Consequently, studies have established that most unsafe abortions [used interchangeably with less safe abortions] take place in sub-Saharan African countries, as women in these countries resort to the use of covert, risky, unorthodox methods, and unsafe means to induce abortion, thereby significantly increasing their risk of dying through complications [1, 2, 10].

The risk associated with unsafe abortion cannot be overemphasized [10–15]. For instance, it has been estimated that unsafe abortion is responsible for an estimated 100–200,000 annual deaths among women in developing countries [2, 4, 13]. According to the WHO, while about 5 million are hospitalized each year for unsafe induced abortion practices, 47,000 lose their lives [2]. Ghana's progress on this issue has been described as slow because unsafe abortions still remain a major public health problem in the country and are currently the second-most contributor to maternal mortality [10]. In 2017, it was estimated that 20% to 27% of all pregnancies in Ghana ended up in abortion [16, 17]. However, the proportion of unsafe abortion practices is very high as 15% of women and girls between the ages of 15–49 years have practiced unsafe abortion [16]. Also, estimated 5% of every 1000 Ghanaian women between the ages of 15–49 years experienced post illegal abortion complication care in 2017. With unintended pregnancies according for half of pregnancies in the country [16], there is the need for policymakers to prioritize public education on sexual and reproductive health.

In Ghana, the law makes provision for safe abortion under the following circumstances: rape, incest, fetal abnormality and individuals who are unable to consent to sexual relationship

[16]. Indeed, safe abortion services are available in both public and private health facilities; however, most members of society are ignorant of this and do not utilize the service. This has culminated in discussions on ways to reduce maternal mortality by creating awareness among women about safe and unsafe abortion practices [18–20]. Indeed, one of the targets of Sustainable Development Goal 3 is to reduce maternal mortality by promoting equitable access to reproductive health services [21–24]. However, deliberations on access to and awareness of safe abortion practices have yet to include deaf women. Some scholarly attention has been paid to the accessibility of reproductive health services for women with disabilities generally [25–31]. However, the knowledge of safe abortion services among deaf women and girls in Ghana is understudied.

## Deafness and access to health care

Deafness is a sensory disability characterized by partial or total hearing loss, which interferes with the reception and processing of auditory information [32]. Thus, deaf individuals are generally unable to hear and use spoken language. Instead, they rely mainly on sign language to communicate [32–35]. According to the WHO, 5% of the global population, estimated at 430 million people, are believed to be living with some form of hearing loss—a figure projected to increase to at least 700 million by 2050 [35]. In Ghana, deaf persons comprise 4% of the over 2 million persons with disabilities [36]. The sizable number of deaf persons underscores the need to include consideration of their basic needs in national development and planning [37–39]. This need has been reiterated in the Convention on the Rights of Persons with Disabilities, which is a global call for countries to mainstream disability issues in development policies such as access to health care [37]. Unfortunately, all over the world, communication barriers have resulted in the exclusion of deaf persons from essential services, such as education, reproductive healthcare, and transportation [32–35].

There are different ways of perceiving deafness, and this variance has implications for the provision and use of services for deaf people. Traditionally, deafness has been perceived as evil and linked to supernatural causes. This belief, which is prevalent in sub-Saharan African countries such as Ghana, sees deafness as a curse or punishment for sins committed by a family member [31, 38–43]. This understanding of deafness associates the birth of a child with deafness to a sin or punishment by a supernatural being, such as the ancestors or gods. This stigmatizes the child and the family. For this reason, the first step often taken by the family is consultation with spiritualists to understand the reason for the birth of the child and to obtain a possible remedy [43].

This traditional interpretation of deafness affects the inclusion of deaf people in larger society [43]. This perception is also responsible for the barriers they encounter in their efforts to access jobs, education, and healthcare [32–35, 37]. Another explanation of deafness sees the condition as a socio-cultural phenomenon. Deaf people, under this conception, see themselves as a linguistic cultural group with a unique language and distinct cultural values that are different from those held by hearing people. Deafness is, therefore, not seen as a disability, pathology, or impairment but as a socio-cultural characteristic [44, 45]. These varied perspectives of deafness suggest that deafness is a complex phenomenon and that the deaf community is not a homogenous entity [46]. The provision of services must therefore consider the distinct features of deaf people, and services must be customized to suit them.

Previous studies that have been conducted to understand the global accessibility of reproductive health services to persons with disabilities have revealed a lack of access [47–53]. In a Cambodian qualitative study, Gartrell et al. studied the reproductive health rights of women with disabilities [52]. They identified physical restrictions, limited access to information,

communication barriers, and financial constraints as factors limiting access. In another qualitative study in Senegal, which assessed the reproductive health needs of young persons with disabilities, Burke et al. reported that financial and attitudinal challenges, as well as the unavailability of health facilities, restrict access to services [48]. In Ghana, comparable challenges have been identified as hindering access to sexual and reproductive health services for women with visual impairments [47] and deafness [38–40]. These included limited access to information, financial constraints, architectural barriers, negative attitudes, and communication barriers. Unfortunately, little is known about deaf women and girls regarding their attitude towards safe abortion practices.

## Utilization of safe abortion services

Studies on the usage of safe abortion services [5–9, 12, 13, 15, 27, 54–63] provide a useful framework within which to situate this study. For example, studies conducted in other developing countries have reported the unavailability of health services to provide safe abortion services to women [58]. In some situations, though health professionals may be available to provide services, and knowledge about safe abortion services may be high among women [58], criminalization of abortion makes it difficult for young women to access services. Moreover, it has been found that the lack of health facilities in some communities, especially in rural areas, negatively impacts women's ability to access abortion-related services [58]. Studies have also been conducted to assess knowledge about and use of safe abortion services in Ghana, and the findings have indicated low knowledge and use. For example, a study conducted in 2007 indicated that only 4% of the women in the study knew that safe abortion was allowed under certain circumstances [55]. Another study, which examined the attitudes of physicians toward safe abortion, revealed that just over half [54%] knew that abortion is legal on health grounds [56].

Furthermore, the intersection of culture and religion, which is partially responsible for negative attitudes among healthcare providers, negatively impacts health-seeking behavior [58]. Even in advanced countries, abortion-related stigma is found to be high among women who have strong religious affiliations [60]. Additionally, factors such as illiteracy, poverty, and low income have been found to hinder access to abortion services in those countries [60, 62]. More directly to this study, obstacles such as illiteracy and communication barriers have been found to have negative impacts on abortion-related services [9–13, 15]. It is evident that complex factors have negative ramifications for access to abortion-related services for women generally. However, scholarly studies on the use of safe abortion services have yet to include the experiences of deaf women.

The deaf community is close-knit, and males and females in the community generally support each other. Due to the stigmatization and isolation of deaf persons in society, deaf males and females are more likely to intermarry [42]. Due to this interconnectedness and intermarriage, we reasoned that this exploratory study on access to safe abortion services ought to include both males and females. Therefore, this study sought to gather the views of deaf persons regarding the knowledge and attitudes towards safe abortion services in Ghana. The study has the potential to provide information about the perception of deaf persons in terms of utilization of safe abortion services.

## Theoretical framework

The multifaceted, encompassing, and interrelated challenges that women face in accessing abortion-related services have been well documented in the literature [8–15, 18]; a broad theory is required to explain these factors in relation to the situation among deaf women and girls. In view of this, Penchansky and Thomas' accessibility to health care theory [63] which is

a multifaceted lens, was chosen to guide this study. Access to health is a common good that ought to be enjoyed by all. However, several factors or determinants, such as beliefs, systems, and human resources, are likely to impact health access [63, 64]. To understand health-seeking behavior, it is essential to consider the relationship between the consumer, the product, and the system [63]. In this study, the consumer is deaf women, whose condition likely places them in disadvantageous positions. Subsequently, the health product here is safe abortion services, which have deemed to be useful by WHO [2]; these should be guaranteed by Ghana's health system. Although the theory of access has received criticism for being limited [65], it has been widely used to study health access, both among the general population and among vulnerable groups [64, 66]. The theory is made up of five interrelated components: availability, accessibility, acceptability, affordability, and adequacy/accommodation [63, 64].

In this study, *availability* is defined as the sufficiency of services related to safe abortion practices. The *demand* for safe abortion services by deaf women will depend on the service *supply* in the community. This will, however, depend on the availability of information available to deaf women and girls. A*ccessibility* denotes services located within the proximate location that can easily be accessed by deaf women and girls. If the facilities providing safe abortion services are conveniently located, deaf women and girls will be encouraged to utilize the services. A*cceptability* refers to the perception of abortion-related services by deaf women. This may be influenced by such factors as cultural understandings of abortion, religious beliefs, and the legal environments and their impacts on the accessibility of abortion services in Ghana. The fourth component, *affordability*, encapsulates the ability of deaf women to cover the costs associated with safe abortions and related services. Finally, *adequacy* and accommodation is the ability of the healthcare system to provide quality health services that meet the unique needs of deaf women and girls seeking safe abortion services. To the best of the authors' knowledge, the theory of accessibility to health care has not been used to assess the perception of deaf women in relation to access to abortion-related services.

## Methods

A phenomenological design guided the conduct of this qualitative study. More specifically, regarding the understanding and use of safe abortion services by deaf women in Ghana, the use of this design enabled the study participants to share rich experiences [67] about perception towards safe abortion services by deaf women in Ghana.

### Study participants

Participants who took part in this study were deaf persons recruited from three of the 16 regions of Ghana the Ahafo region, the Greater Accra region, and the Northern region. The regions were purposely selected to reflect the three geographical divisions within the country: northern belt Northern region, middle belt Ahafo region, and southern belt Greater Accra region. From each region, two districts, the first and last on the list, were randomly selected for data collection. Each district hosts a branch of the Ghana National Association for the Deaf GNAD, which represents the "mother" organization. The GNAD branches advocate for better living conditions for their members at the local level, and the executives at the local level liaise with the national executives to lobby for better conditions for deaf persons in Ghana. In this study, executives at the national level selected the regions and districts where the data were collected. The inclusion criteria were as follows: a membership of GNAD; b resident in the study area; c adult who is 18 years or above; d able to consent to participate in this study. Conversely, prospective participants who did not meet the above criteria were excluded from this study.

In all six selected districts, the executives were able to organize 360 participants who were at least 18 years to take part in this study 120 participants from each of the three districts. The research team sent text and WhatsApp messages to all participants. There was consensus among the research team that the first 10 participants regardless of gender from each district who responded positively would be considered for the study. In each district, one focus group discussion FDG with seven participants and three face-to-face interviews were conducted. GNAD approved FGD for males and females because of the close-knit relationship between members of the deaf community see S1 Appendix. The decision to include both males and females was based triangulation, to compare responses across gender [68]. Similarly, the use of face-to-face interviews and FDG helped to ascertain whether the participants will differ from each other in order to strengthen the findings reported in this study [68].

## Instrument

In this study, qualitative data method was deemed appropriate to gather data about the views and knowledge of deaf persons regarding safe abortion services. As the study focus is a gray area in the literature, the qualitative method was deemed suitable for this study [69]. Specifically, it gave the participants the opportunity to delve deeper into their experiences and share valuable insights that could serve as baseline information. A semi-structured interview guide was used for data collection. The unstructured nature of the tool enabled probing for details from the participants. A review of the literature [4–12, 70–74] and theoretical framework [63] helped us to identify key issues or questions to ask the participants.

## Procedure

The study and its protocols were approved by the ethics review committee at Kwame Nkrumah University of Science and Technology, Ghana CHRPE/AP/375/16. Further approvals were granted from the Ghana Health Service and GNAD before implementation. Subsequently, the GNAD National Office advised on areas where data could be collected. The research team was interested in getting heterogeneous participants to take part in this study, which was communicated to GNAD. In response, the executives identified possible regions and districts in which recruitment could be performed. This was discussed with the research team, and a consensus was reached on the selected districts in each region. GNAD then sent a formal letter to their regional and district representatives, who subsequently invited their members to be part of this study.

A list of potential participants was given to the research team, who invited participation via texts and video WhatsApp messages. The video was used to provide a sign language explanation of the study to any participants who might not have been able to read. The research team decided to consider the first 10 participants from each district who responded to the invitation.

Two trained graduate students who are proficient in Ghanaian Sign Language and English were recruited to collect data from the study participants. In each district, the first seven participants were considered for FDG, while the last three were considered for face-to-face interviews. The data were collected at the GNAD district offices on weekends. While second author was the facilitator for all the FDG, two research assistants, proficient in Ghanaian Sign Language supported data collection—one research assistant was videotaping while another research assistant was taking note. Also, the face-to-face interviews were conducted by the second author.

Before recruitment, all participants were informed that they had the right to withdraw from the study at any time without consequences. They were assured that neither their names nor

any identifying information would be disclosed to anyone outside the research team. The participants were provided with breakfast, lunch, and reimbursement for their transportation to the data collection venue. The interaction was videotaped with permission from the participants, and all the study participants provided written consent before taking part in the study. The use of videotapes has been recommended as an effective approach for gathering rich and accurate data from deaf persons [75, 76]. The recorded interviews and FGDs have been stored on Microsoft OneDrive storage file, purchase for the project and only accessible to the research team. The data will be deleted five years after completion of the study. The interviews and FGD were conducted between June 2017 and September 2020, with a duration ranging from 40 minutes to 2 hours.

## Data analysis

The videotaped data were transferred to a computer for transcription by two research assistants. However, for the sake of accuracy [68], the second author also transcribed one FDG and a face-to-face interview. A meeting was held between the three to compare transcriptions. There was 67% inter-ratter agreement between the three, and consensus was reached on areas where they disagreed. The two research assistants continued to transcribe the videotape. Afterward, the data were shared with some of the participants, who provided feedback, and iterations were suggested where necessary. Additionally, video WhatsApp calls were placed to some other study participants for discussion of the points emerging from the interviews. They were all satisfied with the information and agreed to its use in the reporting.

The data were then subjected to thematic analysis. Since the study was guided by a theoretical framework [77], the components were used as *a priori themes* to guide the analysis. The stages involved in the data analysis were as follows: reading and coding, sorting and mapping, categorization, thematizing, and summary of the results [77]. In the first stage, authors one, two, and three read the transcribed data several times and shared major points that emerged in the data. The next stage was coding the transcribed data. After coding one FGD and interview, they met to discuss the descriptors used for the coding. Where there were disagreements between them, they discussed and reached consensus before continuing with coding. The second stage was sorting and mapping. The authors used sub-themes to sort out relevant information from the data. This enabled the authors to map common ideas and identify areas of disagreement among the participants. The next stage was categorization, and this involved the authors grouping the sub-themes, which were subsequently charted under the a priori themes. Quotations explaining the categories were transferred to a new file. Author three wrote the results section, which was shared with all the authors for their inputs and feedback.

**Positionality and reflexivity.** The reflexivity of the authors stemmed from many years of experience conducting extensive research on disability issues in the study context. This enabled the research team to be aware of possible biases [78]. Also, the research team is aware of the context, practices and culture which enabled accurate reporting of the findings. The research team are key promoters of social model of disability where society is expected to include individuals with disability in all aspect of society including reproductive health. The research team believe that persons with disabilities including deaf persons are entitled to reproductive rights which needs to be promoted and considered in health policies.

Furthermore, two authors one and two of the five co-authors are deaf and members of GNAD, and according to Oliver [79], the involvement of researchers with disabilities in disability research is vital to capturing voices from the field. The diversity of the team enabled the research participants to identify with the research participants who, in our view, provided accurate and useful responses. In the reporting of the study findings, the diversity of the

research team enabled the reporting of accurate findings, which reflected the participants' voices. Discussion and researchers' consensus at each stage of the data analysis were key to reporting accurate data.

## Results

In all, 60 participants took part in this study—six FDG n = 42 and 18 face-to-face interviews were conducted for details, see Table 1. The participants shared their perspectives on knowledge of abortion, access to services related to abortion, causes of abortion, abortion among deaf women and girls, and the use of legal/safe abortion procedures. The information shared by the participants is presented below under the following a priori themes: availability, accessibility, accommodation/adequacy, affordability, and acceptability.

### Availability

Almost all the participants had some knowledge of abortion. Most had heard of abortion through friends, family members, books, health professionals, and the media. However, there

**Table 1. Demographic characteristics of study participants.**

| Categories | Sample N = 60 | % |
|---|---|---|
| **Mode of participation** | | |
| Focus group | 42 | 70% |
| One-on-one interview | 18 | 30% |
| **Gender** | | |
| Male | 18 | 30% |
| Female | 42 | 70% |
| **Age** | | |
| 18–25 years | 33 | 55% |
| 26–35 years | 10 | 17% |
| 36–45 years | 11 | 18% |
| 46 years and older | 6 | 10% |
| **Religion** | | |
| Christian | 34 | 57% |
| Muslims | 22 | 36% |
| Other | 4 | 7% |
| **Educational level** | | |
| Primary level | 31 | 52% |
| High school level | 16 | 27% |
| Tertiary qualification | 13 | 22% |
| **Employment** | | |
| None | 26 | 43% |
| Student | 5 | 8% |
| Self-employed | 19 | 32% |
| Public service | 10 | 17% |
| **Marital Status** | | |
| Single | 26 | 43% |
| Married | 29 | 48% |
| Divorced | 5 | 8% |

**participants were above 18 years. Tertiary = Post-secondary education in Ghana

were disagreements on whether women or young women were engaged in more abortions than were girls. An adult male in an interview stated, "Most of the deaf girls do not want to die from abortion; therefore, they do not engage in it [abortion] often, but the deaf adult women engage in it more than the young girls" Male, Focus Group Participant 1, District B. On the contrary, some participants thought that young girls engaged in abortion more often than did older women because the former were more sexually active. A participant who supported this view reported: "Because the sexual drive is high [among young deaf girls], abortion is more frequent among young girls than old women who have a lower appetite for sex" Male, Focus Group Participant 6, District B.

Regarding knowledge of legal abortion safe abortion, participants mentioned that deaf people were not aware of the law. That is, they did not know that abortion could be legally performed in some situations. This is exemplified in the following statement by an adult male interview participant: "I do not know anything about legal abortion in Ghana. I also think that the deaf people in this area have not heard about legal abortion in Ghana before" Male, Interview Participant 2, District F. Two other interview participants supported this claim:

> In this area, I do not think other deaf people have heard about it [legal abortion]. If a deaf person has heard about it, it is likely that every deaf person in the country would have heard about it, but that is not so among the deaf. We don't know about it. Male, Interview Participant 2, District C

> We know that people have engaged in abortion, but we have not heard about legal abortion. I do not think any deaf person has heard about it. Maybe the hearing people have heard a lot about it, but the deaf people know nothing about it. Female, Interview Participant 1, District C

Regarding whether any deaf person had requested a legal abortion, participants said that, since they did not know much about the laws on safe abortion, they had not requested it. An adult male in an interview explained:

> Legal abortion is not known by many deaf people across the country, so I think it is uncommon for a deaf woman or girl to request it. I also think many of us do not know what legal abortion is, and since we do not know about it, we will find it hard to request it. Male, Interview Participant 1, District E

> I have never heard of any deaf woman or girl requesting it [safe abortion]. This is because legal abortion is unknown to many deaf people [in Ghana], so I think it will be hard for them to request it. If one does not know about it, one would not request it. Female, Focus Group Participant 5, District B

> Some young ladies do not know much about safe abortion, so they go for unsafe abortion to free themselves, and so we need some education and training for them to be aware of it. They need to be educated and shown how and where to get the safe abortion methods from so that they can use them safely. Female, Focus Group Participant 7, District A

Conversely, a few participants said they knew of safe abortion and wanted to request it but could not do so because of financial difficulties: "Yes, I know it, and I desired to have a safe abortion, but I have financial problems, and abortion costs are expensive; we deaf mothers mostly do not work" Female, Focus Group Participant 3, District F.

## Accessibility

Almost all of the participants indicated the availability of health facilities in their communities that are accessible to them. To them, getting to the facility to access abortion services is not a problem; however, they recounted other possible factors that may influence the decision of deaf women to adopt unsafe abortion practices. Some of the most common factors mentioned were communication barriers because of a lack of interpreters, the fear that parents would be angry with them for being impregnated, the fear that they would be judged for getting pregnant before marriage, peer influence, and financial difficulties. For example, a young male in the interview reiterated this view by saying, "Some do it through fear of their family members. They fear that their parents will be angry with them for getting pregnant, so they cause abortion. This is especially for the young ones" Male, Interview Participant 1, District E.

Peer influence has also been identified as a reason for unsafe abortions among deaf women and girls. According to some of the participants, deaf persons are easily swayed by the views of the people they trust. A participant explained: "Some deaf people do abortion because they are being influenced by their peers to do it. Deaf people like following or listening to what their peers say, and this influences them to choose to have an abortion" Female, Focus Group Participant 5, District E. Another participant recounted: "We trust easily. In the general community, no one can communicate with us. We believe whatever those who can communicate with us tell us" Female, Focus Group Participant 7, District A.

Other participants indicated that the fear of not being able to care for a child drives deaf women to unsafe abortion. According to most participants, they struggle to care for themselves, which makes it hard for deaf women to have babies and compound their situation. One participant said, ". . .lack of cash to care for pregnancy-related expenses and the raising of a child alone will be difficult for me. If I cannot care for a child, I have to abort it" Female, Focus Group Participant 4, District E. A female participant added that "men would promise to take care of you. You won't see them again when you get pregnant. The best solution is aborting it to be free" Female, Focus Group Participant 4, District D.

## Accommodation/Adequacy

The participants discussed the difficulties faced by deaf women when they attempt to access abortion services. The major barriers included the stigmas attached to deafness and abortion and a lack of information on the subject in formats that are accessible to deaf people. Most participants indicated that being deaf affects their ability to access essential services. In particular, they cited the hostilities they encountered in their attempts to access health care. For instance, one participant said, "It is difficult to be deaf in Ghana. I'll say that everyone is against you" Male, Focus Group Participant 1, District B Another person said, "The nurses and doctors don't respect us. I don't think we will go to them for such information. They ignore me when I go there." Female, Focus Group Participant 2, District C

Participants further shared their thoughts on the stigma attached to abortion, which inhibits access. An adult male participant explained the challenges deaf people encounter when attempting to access services related to abortion: "It is difficult because most deaf people think it is bad to ask for information and services on abortion because it [abortion] is a negative thing" Male, Focus Group Participant 5, District C. Supporting this view, a female interview participant said that "most of us, especially the women, feel shy to talk about abortion because we are not encouraged to ask for information and services on abortion" Female, Focus Group Participant 6, District D. A male interview participant attributed the problem to communication barriers: "It is difficult to get information due to lack of interpreters and communication problems, and information on abortion on the TV is inaccessible" Male, Focus Group

Participant 2, District F. Two other participants recounted the challenges that deaf people face in accessing information on abortion:

> Abortion is not commonly practiced among deaf women; therefore, to inquire for information and services on it is very difficult. This is because it is not easy to find someone who knows more about it to tell you about it. Other sources of information, such as TV and radio stations, are also not accessible to deaf people. Even if I have an interpreter, it will be hard because I will be ashamed to talk about it. Female, Focus Group Participant 4, District F

> It is very difficult for the deaf to access abortion information because of fear of being misunderstood by health workers and other people. We lack interpreters to help us with translating what the doctors offer in their services since we cannot hear them speak orally to us, and this makes it difficult for deaf people to access information on abortion. Female, Focus Group Participant 2, District C

It is clear that, because of these barriers and the lack of knowledge about safe abortion procedures, some deaf women and girls resort to the use of unsafe abortion procedures. Some of the methods identified during interaction with the participants were taking pills, drinking concoctions, such as soft drinks mixed with sugar and herbs, and taking pain killers. Two participants described some of the methods and why they used them.

> They [deaf women and girls] use pills [for abortion] . . . they do not use qualified doctors because communication is difficult between the deaf and doctors. Some also abuse painkillers to abort the baby. These are dangerous, but because of communication barriers, deaf people engage in these practices. Male, Interview Participant 2, District A

> Some buy soft drinks, add lots of sugar, and drink it to abort the baby. Some use herbal concoctions, with no qualified doctor involvement. Boiling some various herbs and drinking them can cause abortion, but I have not heard of any deaf person involving qualified doctors. Maybe because of communication issues and being shy.Male, Focus Group Participant 6, District B

## Affordability

Many of the participants knew of ways to prevent unwanted pregnancies. The methods mentioned by the participants were abstinence, using birth control pills, condoms, or injectables, and following one's menstrual cycle. Participants indicated that some deaf people knew about prevention methods and where they could obtain abortion services. However, it was agreed that not all deaf people knew about these methods.

> Some young ladies do not know much about methods of preventing unwanted pregnancies, so they go for unsafe abortions to free themselves, and so we need some education and training. They need to be educated and to be shown how and where to get safe abortions so that they can use them safely. Female, Focus Group Participant 5, District E

> We are aware of such methods [abortion prevention methods], but some do have difficulty attending the health centers where the services are provided for advice because they fear that their issues will not be kept confidential. That is, they fear the people working there [health workers] will tell others about what the women came to do. Male, Interview Participant 2, District F

They [deaf women and girls] are aware, but they do not have money to visit the hospitals because it is expensive. Many deaf women do not have jobs to get money for an abortion. Also, the men who made them pregnant do not care. Since they are not able to afford to go to the hospital, they use the wrong methods. Male, Interview Participant 3, District A

The major concern raised by almost all participants was that safe abortion services were very expensive, which contributed to deaf women's inability to access such services. Most participants noted that safe abortion services have not received much publicity, and as such, some health professionals use them as a way to make money. One participant indicated that "the doctors make a lot of money from it. I'm not working and can't pay for it" Female, Interview Participant 2, District E. Another participant indicated, "It is one of the most expensive services. We don't talk about it, so people [health workers] see it as a way to generate extra income" Male, Focus Group Participant 1, District B. Supporting this, another participant asserted that "most deaf women are not working and can't afford it. It's an expensive service" Male, Interview Participant 1, District E.

## Acceptability

There were also divergent views on whether abortion should be legalized. While some of the participants thought abortion should not be legal in any circumstances, others thought it should be allowed under certain conditions. Those who disagreed with legalizing abortion thought abortion could be harmful to the woman and that it was immoral to engage in the practice. They also expressed their unwillingness to recommend it to other deaf women due to religious reasons and the likelihood that people would begin to engage in more unprotected sex. One of the participants who was against abortion explained that "abortion should not be allowed under any conditions. It is a bad thing that can harm the woman, and also, it is a sin to take someone's life A male interview Participant. It is "a wicked act of killing babies; therefore, it is not a valuable option for women" Male, Focus Group Participant 6, District B. A speaker in the focus group explained:

No, I will not recommend abortion as an option under any circumstance because it is against my religion. Also, I will not because many young people will engage in sex. They will engage in sex without protecting themselves. For instance, many would have sex without the use of a condom because they know they can have a [safe] abortion if anything went wrong. Female, Interview Participant 2, District E

Participants who supported abortion thought it should be permitted when the life of the woman is in danger, but it should be done by a professional. Some participants said they would recommend safe abortion as an option to their relatives and neighbors if it were the only way to save their lives. A focus group participant noted this in response to a question that asked if she would recommend abortion to someone else: "No, I will not [recommend] but will do so only when the person suffers or experiences heavy bleeding and her life is in danger" Female, Focus Group Participant 5, District B. This view was supported by another focus group participant who said, "Yes, I will if the woman has a condition that would make her die" Female, Focus Group Participant 4, District D. Below are excerpts from three participants:

Abortion should be allowed if there is a reason for it, and it is also safe. If the mother's life is in danger and abortion is the only remedy by which the mother's life would be saved, then it should be done with the help of a professional doctor. Male, Focus Group Participant 6, District B

Abortion can be good or bad. If there is any justifiable reason behind it, it should be allowed. For example, when the mother's health is affected by a pregnancy of which abortion is necessary, it should be allowed to save the mother. Male, Focus Group Participant 3, District C

If the woman would be affected badly in her health and safe abortion is the only remedy that can save her from dying, then I think the law should allow it [abortion]. Also, I will allow my wife to do abortion only if her health is affected badly and abortion is the only remedy. Male, Interview Participant 1, District E

Others said they would recommend abortion if the pregnancy is the result of rape or if the victim involved is a student: "Yes, I will recommend abortion to people if either the pregnant girl or woman being raped still needs to continue school" Female, Focus Group Participant 5, District A. One participant claimed that it could also be used for birth control.

Many women are dying as a result of the wrong methods of abortion they used, so I think if there is such a thing as safe abortion, then it is good. It will allow women to seek a safe one without fear. Abortion could also be used as a family planning method, which is good because people can use safe abortions to reduce the number of children they give birth to. Female, Focus Group Participant 4, District B

## Discussion

In this study, we aimed to understand opinions on of the usage of safe abortion services among deaf women and girls in Ghana. It was evident that at all levels of the theory, there were barriers that restricted the use of safe abortion services by deaf women. In relation to accommodation, the results showed a double stigma faced by deaf women when accessing services. Deafness is considered an "unacceptable" condition by most people [38, 40], and this unacceptability intersects with abortion, which is highly sensitive and taboo in Ghanaian society. Those who engage in the practice are stigmatized [10, 13, 80]. Consequently, deaf women may be reluctant to initiate a conversation about abortion, preferring instead to avoid the consequences of engaging in a practice that is deemed a "sacrilege" in Ghanaian society [12, 13]. This could put the lives of deaf women at risk, as they may be pushed to adopt unsafe abortion practices.

The study found that financial barriers significantly restrict the utilization of safe abortion services by deaf women. Deaf women who defy the stigma to access safe abortion services may spend a lot on the service. This cost includes employing the services of trusted sign language interpreters and the cost of the abortion itself. However, there is evidence indicating that women with disabilities are among the poorest of the poor [81]. Compared to men with disabilities, women with disabilities are less likely to have access to education and jobs [81]. There is also evidence that most deaf women are unemployed, as they do not have the necessary qualifications or skills to access jobs [40]. This suggests that deaf women may be less likely to have the financial resources needed to cover the expense of a safe abortion. Additionally, deaf women and girls are at increased risk of rape and other sexual abuses, putting them at a high risk of unwanted pregnancies that would require an abortion [81–84]. With no help from the father [82], limited access to financial resources [44, 83], and no social protection for individuals with disabilities in Ghana [85, 86], they may be compelled to engage in unsafe abortion practices.

Religious affiliations can be another barrier to the use of abortion services. In the Ghanaian traditional society, procreation is considered a blessing [87, 88]. In fact, a person is socially measured based on their assets and the number of children they have. Christianity and Islam —most Ghanaians belong to one or the other—do not support abortion. Consequently,

abortion is strongly abhorred and rejected by many Ghanaians, and this seems to have influenced the perceptions of the study participants. Due to the cultural stereotype around abortion, there is no discussion of safe abortion practices. This has probably made it difficult for participants to distinguish between safe and unsafe abortion practices as both terms have been placed under one umbrella. This finding is consistent with previous studies that reported the rejection of abortion by society because of religion and local culture [7, 10, 13]. However, there was a consensus that a safe abortion may be performed if the life of the mother is at risk. This suggests the value that the participants and, for that matter, Ghanaian society put on human lives; abortion is rejected on the basis that it amounts to killing, but it can be accepted if the life of the mother is at stake. This thinking is somewhat in line with the global attention that is currently being paid to methods of reducing maternal mortality [14]. Thus, policymakers could concentrate on educating deaf women about safe and unsafe abortion practices, the conditions under which abortion it is permissible and avenues to access services. However, awareness creation should be conducted alongside the removal of barriers to safe abortion services.

Another accessibility issue is the inability of health workers to communicate with deaf women. Communication is fundamental in day-to-day living activities, and it is particularly important in healthcare settings. The ability of individuals to assert their rights and demand services from their society is dependent on their ability to communicate effectively with others [32]. However, the results showed a communication barrier between deaf women and health workers. This finding is unsurprising, as the literature has reported the inability of health workers to communicate with deaf persons [25, 26, 89, 90]. Closely related to the language barrier is the lack of privacy for deaf women seeking abortion services. Privacy is fundamental to access to health care among deaf women and other vulnerable groups; anyone seeking services, such as those related to abortion, must be assured of their privacy. Abortion is highly stigmatized and culturally unacceptable in Ghana [12, 90, 91], resulting in serious accessibility barriers for deaf women and girls. While sign interpreters are required to break the communication barrier with healthcare workers, the presence of an interpreter could breach the privacy of deaf women and force them to withhold vital information. That is, some women would feel uncomfortable sharing or discussing issues in the presence of the interpreter. This is the paradox that deaf women and girls face when accessing sexual and productive health services.

This finding of a lack of privacy for deaf women and girls is consistent with a previous study that identified a lack of privacy as a reason for the non-utilization of healthcare services [37]. This finding is also consistent with other studies that found a lack of privacy among hearing women to be a major issue in their decision not to seek safe abortion services [10]. Privacy is therefore a serious accessibility issue that should be addressed when providing safe abortion services for deaf people. Although there are laws regulating safe abortion practices in Ghana, it was apparent that most participants in this study were unaware of them. The unfamiliarity of laws related to safe abortion appears to be a common phenomenon in Ghana, as studies on women [55] and health practitioners [56] have reported a similar lack of knowledge. Although low levels of formal education and communication barriers are possible reasons for this limited knowledge, the same cannot be said of participants in other studies. There seems to be a general lack of public education on the topic in Ghana. Instead, emphasis seems to have been placed on the criminal aspects of the law; abortion is often perceived as a crime in Ghana, and little has been done to educate the public on non-criminal, safe abortion procedures.

## Study limitations

The findings of this study are not generalizable because of a number of limitations. First, all the participants were deaf people who used Ghanaian Sign Language. There are many deaf

persons in the country who do not use Ghanaian Sign Language, and these people do not benefit from interpretation services. Instead, they have unique communication systems that only the people they live with understand. Their experiences using abortion services may be different, but they could not be captured in the current study. Second, there was no consideration given to whether participants had other comorbidity conditions that could influence access to services. Third, there is potential for study bias, as GNAD provided input regarding the study site and the recruitment of potential participants. It is possible that the executives invited only individuals who might provide certain responses. However, decisions about who would be included in the interviews were made by the research team, which devised a means of identifying participants from the pool. Four, the study was conducted between 2017–2020 and since safe abortion is sociocultural factor, there is possibility for change in perception of study participants. Five, power dynamics between male and female participant could influence responses of study participants who participated in the mixed group FGD. This limitation was noted during the data collection stage which led to the consultation with GNAD. It was clarified that deaf women are more "outgoing, outspoken and domineering" than deaf men. Thus, the possibility of power dynamic might not exist within the community compared to the general population. Future studies could use a quantitative design to understand the background variables that may influence the understanding of deaf persons regarding access to abortion services. Overall, a major strength of this study was recruiting male and female deaf persons to share their perspectives on this sensitive topic.

## Implications for policymaking

The findings of this study suggest the need for a multifaceted approach to enhancing the utilization of safe abortion services among deaf women in developing contexts, such as Ghana. First, there is a need for public education to improve awareness about deaf people and their needs, especially among healthcare workers. This would reduce misconceptions and the stigma associated with deafness as well as challenge society to embrace deaf persons as equal members. Awareness will increase understanding of the deaf community and their needs, thereby helping to develop deaf-friendly abortion services. Second, to further progress toward achieving inclusive healthcare, health policymakers need to consider the communication needs of deaf persons. One approach could be to train health professionals in sign language, thereby bridging the communication gap without breaching the privacy of deaf clients. Third, there is a need for health policymakers to collaborate with bodies such as GNAD when conducting public education on safe abortion practices. This would help GNAD acquire the necessary information on safe abortion and incorporate it into its advocacy efforts. Fourth, financial assistance could be provided to support disabled women who need safe abortion services. Since the lack of awareness of safe abortion practices is a general problem in Ghana, the Ghana Health Service should intensify public education on these services to improve awareness among the general public and reduce the stigma associated with abortion.

## Conclusion

The purpose of this study, which used Penchansky and Thomas' [63] accessibility to health theory as a guide, was to collect data and perspective on the use of safe abortion services among deaf women and girls in Ghana. It was found that at every level of the theory, there were barriers to the use of safe abortion services for deaf women. While there are safe abortion services available in most healthcare centers, deaf women were unable to exercise the use of these services and other personal reproductive rights. However, many of these challenges are not particular to the deaf community. Nevertheless, steps that are being made to improve

access for women appear not to have reached deaf women and girls. The findings of this study therefore call for mainstreaming the needs of deaf women and ensuring that they have easy access to safe abortion services.

## Supporting information

**S1 Appendix. Disaggregation of study participants.**
(DOCX)

## Acknowledgments

The authors wish to Ghana National Association for Deaf for their support in the recruitment of participants for this study.

## Author Contributions

**Conceptualization:** Wisdom Kwadwo Mprah, Juventus Duorinaah, Maxwell Peprah Opoku, William Nketsia, Michael Amponteng.

**Data curation:** Wisdom Kwadwo Mprah, Juventus Duorinaah, Maxwell Peprah Opoku, William Nketsia.

**Formal analysis:** Wisdom Kwadwo Mprah, Juventus Duorinaah, Maxwell Peprah Opoku, William Nketsia.

**Investigation:** Wisdom Kwadwo Mprah, Juventus Duorinaah, Maxwell Peprah Opoku, Michael Amponteng.

**Methodology:** Wisdom Kwadwo Mprah, Maxwell Peprah Opoku, William Nketsia, Michael Amponteng.

**Project administration:** Juventus Duorinaah.

**Writing – original draft:** Wisdom Kwadwo Mprah, Juventus Duorinaah, Maxwell Peprah Opoku, William Nketsia, Michael Amponteng.

**Writing – review & editing:** Wisdom Kwadwo Mprah, Juventus Duorinaah, Maxwell Peprah Opoku, William Nketsia, Michael Amponteng.

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
