## [Decision Letter · Decision Letter 0]

1 Aug 2022

PONE-D-22-01224Usage of safe abortion services to deaf women and girls in GhanaPLOS ONE

Dear Dr. Opoku,

Thank you for submitting your manuscript to PLOS ONE. After careful consideration, we feel that it has merit but does not fully meet PLOS ONE’s publication criteria as it currently stands. Therefore, we invite you to submit a revised version of the manuscript that addresses the points raised during the review process.

Please note that we have received comments from one reviewer that have raised major ethical concerns with your study. Please ensure you read these comments carefully and clearly address them in a revised submission. 

We look forward to receiving your revised manuscript.

Kind regards,

Carla Pegoraro

Division Editor

PLOS ONE

Journal Requirements:

Reviewers' comments:

Reviewer's Responses to Questions

**Comments to the Author**

1. Is the manuscript technically sound, and do the data support the conclusions?

Reviewer #1: Partly

2. Has the statistical analysis been performed appropriately and rigorously? 

Reviewer #1: N/A

3. Have the authors made all data underlying the findings in their manuscript fully available?

Reviewer #1: No

4. Is the manuscript presented in an intelligible fashion and written in standard English?

Reviewer #1: Yes

5. Review Comments to the Author

Reviewer #1: General comments

The paper titled “Usage of safe abortion services among deaf women and girls in Ghana” aims to provide an important contribution to the existing literature on abortion in Ghana, particularly on reproductive health care and the rights of vulnerable women. The paper aimed at enhancing understanding of the experiences of deaf women and girls when accessing safe abortion services. To achieve this, semi-structured interviews and focus group discussions were used and guided by the components of the accessibility to health theory by Penchansky and Thomas to collect data for analysis. Findings from the study show gaps in the provision of safe abortion services, especially for vulnerable groups such as women with deafness.

However, there are ethical and several methodological concerns that require explanation and revisions to enhance the quality of the findings. The study used videotaped interviews with permission from the participants. There are usually ethical concerns about videotaped interviews, especially in this case, where audiotape is sufficient. What informed the use of video and not audio? This is troubling, especially on privacy and confidential grounds. The sample used is a vulnerable group and probably coerced for this videotape. Was the videotaping procedure indicated in the protocol approved by Ghana Health Service (GHS)? I also noticed that the ethical approval number from the GHS ethics review committee has not been stated. Kindly incorporate this in the paper.

It is also unclear to me why males were part of the sample, and yet, the title is specifically for women and girls with deafness. The voices of men usually dominate in focus group discussions of both males and females. With this, it could be possible that the findings reflect the opinion of men and not necessarily of women and girls. The sample comprises males and females who did not necessarily induce abortion in their lifetime. The strength of this study rests on experiences and it is unclear how these findings can be trusted, especially when the respondents may not have experienced abortion before. Is not possible that the interviews and focus group discussions reflect what they have heard and necessarily what was experienced? I have provided other comments for consideration based on the sections of the paper.

Abstract

1. The statement “The need for policymakers to expedite public education on acceptance of deaf persons as equal members so society” in the conclusion sub-section requires rephrasing. I do not think people with disability are not considered equal members of society. The problem has to do with the designing and programming of interventions. Most interventions often fail to consider the special need of such groups and eventually these people are lopped out of the benefits associated with the interventions.

Introduction

2. Revise this statement “Induced abortion, which can be safe or unsafe, is a procedure undertaken intentionally to terminate an unintended pregnancy before the fetus is capable of extrauterine life (1)”. WHO has further categorized unsafe abortion as less safe and least safe. Read the article below and revise.

Ganatra B, Gerdts C, Rossier C, Johnson BR, Tuncalp O, et al. Global, regional, and subregional classification of abortions by safety, 2010–14: estimates from a Bayesian hierarchical model. The Lancet.

2017;390(10110):2372–81.

3. This statement “In fact, there is no study on deaf woman and girls regarding safe abortion practices” requires rephrasing. I do not think the authors have explored studies that are not published or probably in universities' repositories.

Study participants

4. How were the districts and regions selected? Kindly be explicit on this. Were the districts/regions selected randomly or purposively?

5. “In all, 60 participants took part in this study – six focus group discussions (n=42) and 18 face-to-face interviews were conducted. (See Table 1 below for details)”. This statement seems like results. Kindly move Table 1 to the Results section.

6. Intervals for the age categories in Table 1 are not the same. Revise for the interval to be the same for the first three categories.

7. “The data collection method was deemed appropriate we attempt to develop insight into the experiences of deaf women in their effort to access safe abortion services”. Not clear. Kindly rephrase this statement.

8. “Since not much has been done in this area, the qualitative method was identified as suitable for this study”. This statement contradicts your earlier statement in the introduction section, where you indicated that nothing has been done in this area.

Procedure

9. “The study and its protocols were approved by the Ghana Health Services before implementation”. Indicate the approval number from Ghana Health Service Ethics Review Committee.

10. “videotaped”. Refer to my comment on the videotaping.

11. “The interviews and focus groups were conducted between June 2017 and September 2020”. Over three years period for interviewing respondents raises the issue of timing. Three years is a lot of time for significant changes in the demographic, social, economic and reproductive attributes of women. For instance, some of the women within the age bracket "15-25" may not be in the same bracket at the end of the 3 years period.

Discussion

12. “we aimed to understand the extent of use of safe abortion services among deaf women and girls in Ghana”. Have the respondents ever used abortion services? So far, it looks like a no answer. In that case, how can we understand the extent of use from a sample that has not experienced abortion or induced abortion?

Additional comment

The paper requires proofreading from an English expert. I noticed several grammatical errors that require to be revised to enhance the flow of reading.

6. PLOS authors have the option to publish the peer review history of their article (what does this mean?). If published, this will include your full peer review and any attached files.

Reviewer #1: **Yes: **Caesar Agula

---

## [Decision Letter · Decision Letter 1]

27 Dec 2022

PONE-D-22-01224R1Usage of safe abortion services among deaf women and girls in GhanaPLOS ONE

Dear Dr. Opoku,

Thank you for submitting your manuscript to PLOS ONE. After careful consideration, we feel that it has merit but does not fully meet PLOS ONE’s publication criteria as it currently stands. Therefore, we invite you to submit a revised version of the manuscript that addresses the points raised during the review process.

The manuscript has been evaluated by three reviewers, and their comments are available below.

The reviewers have raised a number of concerns that need attention. Most of these are requests for additional information and clarification.

Could you please revise the manuscript to carefully address the concerns raised?

We look forward to receiving your revised manuscript.

Kind regards,

Steve Zimmerman, PhD

Associate Editor, PLOS ONE

Reviewers' comments:

Reviewer's Responses to Questions

**Comments to the Author**

1. If the authors have adequately addressed your comments raised in a previous round of review and you feel that this manuscript is now acceptable for publication, you may indicate that here to bypass the “Comments to the Author” section, enter your conflict of interest statement in the “Confidential to Editor” section, and submit your "Accept" recommendation.

Reviewer #2: All comments have been addressed

Reviewer #3: All comments have been addressed

Reviewer #4: All comments have been addressed

2. Is the manuscript technically sound, and do the data support the conclusions?

Reviewer #2: Partly

Reviewer #3: Yes

Reviewer #4: Yes

3. Has the statistical analysis been performed appropriately and rigorously? 

Reviewer #2: Yes

Reviewer #3: N/A

Reviewer #4: Yes

4. Have the authors made all data underlying the findings in their manuscript fully available?

Reviewer #2: No

Reviewer #3: No

Reviewer #4: No

5. Is the manuscript presented in an intelligible fashion and written in standard English?

Reviewer #2: Yes

Reviewer #3: Yes

Reviewer #4: Yes

6. Review Comments to the Author

Reviewer #2: Authors’ efforts are really appreciated but the following comments and recommendations are kindly provided for the stronger presentation about the study.

1. Line number are needed to add according to the PLOS ONE reporting guideline

2. Introduction needs in detail about legal abortion services in the study area. It is also necessary to mention the incidence of unsafe abortion among deaf women and girls.

3. Interconnectedness and intermarriage within the deaf community is not strongly Justified for the present study and to link with the Penchansky and Thomas’ accessibility to health care theory. Therefore, more specific justification is suggested to apply such theory.

4. Detailed inclusion and exclusion criteria for focus group discussion FGD should to be included

5. It is good to mention why both males and females were selected for the same FGD section.

6. It will be better to mention whether menopaused women participated in this study because their interest may be different from that of the younger aged groups.

7. The detailed procedure of FDG should be described in data collection section such as the facilitator role, number of note taker and their role.

8. The aim of face-to-face interview in addition to FGD is also needed to explore.

9. Are there any specific criteria of participants for face-to-face interview?

10. Place of data collection should be mentioned because it is important to keep confidentiality and personal information of the participants related to the abortion services.

11. Author is kindly suggested to mention the timing of face-to-face interview like that of FGD.

12. Working definition of tertiary qualification is suggested to include because readers from different areas and countries cannot know.

13. Is there any information about marital status of the participants obtained during data collection and did the investigator consider the effect of either being married or single status on usage of safe abortion services?

14. Quotation about cost of care are described under the title of availability and accessibility. Therefore, these findings should be analyzed and categorized again according to the working definition in the theoretical framework.

Reviewer #3: This is a very important work. Thank you for addressing all the comments. The article have been written in a very intelligible manner. I wish all the authors good luck with their future work.

Reviewer #4: Thank you for this very interesting paper and important work highlighting the experiences of deaf

women and girls when accessing abortion services. Thank you for addressing the reviewers comments, especially around the ethical implications of videotaping and the reason for its use. The authors have provided a great deal of supporting evidence and socio-cultural context for the study in the introduction, and have used a great theory; Penchansky and Thomas’ accessibility to health care theory, to derive their interview questions and inform their results.

Please find attached some suggested revisions:

Abstract

- “With unsafe abortion being a major cause of maternal deaths among women in developing countries, there is a need to explore the extent to which deaf women and girls in Ghana use safe abortion services” I think this statement needs to be rephrased, as it implies that the aim of the study was to explore the prevalence/how much women and girls in Ghana are accessing safe abortion services and their experience with service delivery, whereas it seems the aim is more around the communities perceptions/attitudes towards safe abortion? The revised paragraph on Page 7, section “Utilization of safe abortion services” does seem to clarify this by saying “Therefore, this study sought to gather the views of deaf persons regarding the accessibility of abortion services in Ghana and deaf women’s knowledge of these services”. I would suggest kindly changing the title of the paper and abstract accordingly.

- “Objective: The main aim of this study was to increase understanding of the experiences of deaf women and girls when accessing safe abortion services.” As above

Introduction

- Page 3 “An induced abortion is a procedure undertaken intentionally to terminate an unintended pregnancy before the fetus is capable of extrauterine life (1)”. Reference 1 is a better reference for the categories of abortion (safe to unsafe) authors rather than a single definition of abortion. I would suggest that this definition should be updated, as some abortions may occur for an intended pregnancy with changing circumstances, and some late term abortions may occur in some countries in certain circumstances including threat to the pregnant person’s life. An updated definition could be from the CDC such as: “a legal induced abortion is defined as an intervention performed by a licensed clinician (e.g., a physician, nurse-midwife, nurse practitioner, physician assistant) within the limits of state regulations, that is intended to terminate a suspected or known ongoing intrauterine pregnancy and that does not result in a live birth” https://www.cdc.gov/reproductivehealth/data_stats/abortion.htm.

- Page 3 “……..as women in these countries resort to the use of clandestine, risky, unorthodox, and unsafe means to induce abortion, thereby significantly increasing their risk of dying through complications (1-3, 9)”. I would suggest here that some different wording could be used? For example “covert” instead of “clandestine and perhaps a different word for “unorthodox” or even the use of “unorthodox methods”? While the words clandestine and unorthodox do indeed refer to secrecy and alternative respectively, these particular words may infer that the act is “socially unacceptable” as opposed to done in secret due to legal restrictions.

Methods

- A reflexivity statement would be great to further explore the researchers positionality and also views and attitudes towards the research topic

Instrument

- Page 10, line 36, We used a qualitative data collection method to gather data that granted insight into the experiences of deaf women in their efforts to access safe abortion services”. This sentence implies that the aim of the study is to interview women who have attempted to access safe abortion services, whereas it seems the aim of the study are more around general accessibility and knowledge. Kindly suggest this sentenced is rephrased, and include the aim of the study as listed above and in the introduction: 1) views of deaf persons regarding the accessibility of abortion services in Ghana 2) deaf women’s knowledge of these services

Procedure

- Thank-you for updating the document to note the reason for using videotaping of interviews/focus groups. Please provide some information on how and where the videotapes will be stored safely to ensure patient confidentiality.

- Page 11, “The interviews and focus groups were conducted between June 2017 and September 2020, with a duration ranging from 40 minutes to 2 hours”. Thank you for addressing the reviewers’ comments re the long duration of data collection. While I acknowledge the process can take years to complete, it may be worth adding this as a limitation of the study in the Discussion – as socio-cultural factors can change over time as noted by the previous reviewer.

Data analysis

- Page 11, “The videotaped data were transferred to a computer for transcription by two research assistants”. Were the research assistants also proficient in sign language? Or did the data collection researchers translate as the focus groups/interviews were conducted? This is an important context to add as it would add an extra level of interpretation of participants views/experiences

Results

- Thank you for making an amendment to the methods to explain why male participants were included, however, I note that the results feature many quotes from male participants in the study. As noted by a previous reviewer, this should be noted as a limitation in the discussion section, as firstly, due to power imbalances between men and women, there is a chance that women in focus groups did not feel able to speak their views.

- If there are quotes from female participants that have undertaken the abortion process it would strengthen the results. I would suggest that due to the nature of participants and results, the study introduction and title should be more framed around the deaf community perceptions rather than being about women and girls experiences.

Discussion

- Page 22, “Consequently, abortion is strongly abhorred and rejected by many Ghanaians, and this seems to have influenced the perceptions of the study participants” This is an important point, and one that highlights the incongruence between some of the results – for example, if someone does not agree with abortion this would certainly have effected their perception of barriers as the authors have noted. In the results section, such as results under “Accessibility” it may be worthwhile noting the beliefs of participants around abortion at the start, and whether the quotes or findings are in relation to a participant that is pro-safe abortion generally, abortion only in certain circumstances, or that abortion should not be legal. Or if not amending the results section, kindly expand in the discussion on how their perceptions have been influenced and examples of this.

Thank you again for the research paper, I look forward to reading a revised manuscript of what could be an excellent addition to the literature.

7. PLOS authors have the option to publish the peer review history of their article (what does this mean?). If published, this will include your full peer review and any attached files.

Reviewer #2: No

Reviewer #3: **Yes: **Raafat Hassan

Reviewer #4: No

---

## [Decision Letter · Decision Letter 2]

7 Feb 2023

Knowledge and Attitudes of Deaf Persons towards Safe Abortion Services in Ghana

PONE-D-22-01224R2

Dear Dr. Opoku,

We’re pleased to inform you that your manuscript has been judged scientifically suitable for publication and will be formally accepted for publication once it meets all outstanding technical requirements.

Kind regards,

Nabeel Al-Yateem, PhD

Academic Editor

PLOS ONE

Additional Editor Comments (optional):

Reviewers' comments:

Reviewer's Responses to Questions

**Comments to the Author**

1. If the authors have adequately addressed your comments raised in a previous round of review and you feel that this manuscript is now acceptable for publication, you may indicate that here to bypass the “Comments to the Author” section, enter your conflict of interest statement in the “Confidential to Editor” section, and submit your "Accept" recommendation.

Reviewer #2: All comments have been addressed

2. Is the manuscript technically sound, and do the data support the conclusions?

Reviewer #2: Yes

3. Has the statistical analysis been performed appropriately and rigorously? 

Reviewer #2: Yes

4. Have the authors made all data underlying the findings in their manuscript fully available?

Reviewer #2: Yes

5. Is the manuscript presented in an intelligible fashion and written in standard English?

Reviewer #2: Yes

6. Review Comments to the Author

Reviewer #2: Well appreciate for your detailed response to each and every comment and question. Hope to be published soon.

7. PLOS authors have the option to publish the peer review history of their article (what does this mean?). If published, this will include your full peer review and any attached files.

Reviewer #2: **Yes: **May Soe Aung, Associate Professor, University of Medicine (1), Yangon

---

## [Editor Report · Acceptance letter]

10 Apr 2023

PONE-D-22-01224R2 

Knowledge and Attitudes of Deaf Persons towards Safe Abortion Services in Ghana 

Dear Dr. Opoku:

I'm pleased to inform you that your manuscript has been deemed suitable for publication in PLOS ONE. Congratulations! Your manuscript is now with our production department. 

Kind regards, 

on behalf of

Dr. Nabeel Al-Yateem 

Academic Editor

PLOS ONE